# Newborn Screening for Duchenne Muscular Dystrophy: First Year Results of a Population-Based Pilot

**DOI:** 10.3390/ijns8040050

**Published:** 2022-09-22

**Authors:** Michael J. Hartnett, Michele A. Lloyd-Puryear, Norma P. Tavakoli, Julia Wynn, Carrie L. Koval-Burt, Dorota Gruber, Tracy Trotter, Michele Caggana, Wendy K. Chung, Niki Armstrong, Amy M. Brower

**Affiliations:** 1American College of Medical Genetics and Genomics (ACMG), Bethesda, MD 20814, USA; 2Division of Genetics, Wadsworth Center, New York State Department of Health, Albany, NY 12208, USA; 3Columbia University Irving Medical Center, New York, NY 10032, USA; 4Department of Pediatrics, Cohen Children’s Medical Center, Northwell Health, New Hyde Park, NY 11040, USA; 5Departments of Pediatrics and Cardiology, Donald and Barbara Zucker School of Medicine at Hofstra/Northwell, Hempstead, NY 11549, USA; 6American Academy of Pediatrics, Itasca, IL 60143, USA; 7Parent Project Muscular Dystrophy, Washington, DC 20005, USA

**Keywords:** newborn screening, Duchenne muscular dystrophy, pilot, dystrophinopathy, recommended uniform screening panel, therapy

## Abstract

Advancements in therapies for Duchenne muscular dystrophy (DMD) have made diagnosis within the newborn period a high priority. We undertook a consortia approach to advance DMD newborn screening in the United States. This manuscript describes the formation of the Duchenne Newborn Screening Consortium, the development of the pilot protocols, data collection tools including parent surveys, and findings from the first year of a two-year pilot. The DMD pilot design is population-based recruitment of infants born in New York State. Data tools were developed to document the analytical and clinical validity of DMD NBS, capture parental attitudes, and collect longitudinal health information for diagnosed newborns. Data visualizations were updated monthly to inform the consortium on enrollment. After 12 months, 15,754 newborns were screened for DMD by the New York State Newborn Screening (NYS NBS) Program. One hundred and forty screened infants had borderline screening results, and sixteen infants were referred for molecular testing. Three male infants were diagnosed with dystrophinopathy. Data from the first year of a two-year NBS pilot for DMD demonstrate the feasibility of NBS for DMD. The consortia approach was found to be a useful model, and the Newborn Screening Translational Research Network’s data tools played a key role in describing the NBS pilot findings and engaging stakeholders.

## 1. Introduction

Advances in therapies for Duchenne muscular dystrophy (DMD) have made DMD diagnosis within the newborn period a priority for the Duchenne community. DMD is an X-linked lethal disorder caused by mutations in the Dystrophin gene. The spectrum of dystrophinopathy includes both Duchenne muscular dystrophy and Becker muscular dystrophy (BMD). Partially functional (in BMD) or absent dystrophin (in DMD) results in destabilization of the sarcolemma, which leads to muscle cell death and replacement by fat and fibrosis. Progression of the disease typically leads to cardiomyopathy and respiratory failure, which are the leading causes of death among individuals with DMD. Newborn screening (NBS) for DMD would facilitate early diagnosis and intervention before the onset of observable clinical disease, and experts believe that earlier treatment improves health outcomes [1,2,3,4,5]. Steroid therapy is used to slow disease progression and improve quality of life. Four exon-skipping therapies have been approved by the Food and Drug Administration (FDA), with others progressing through clinical trials [6]. Additional treatments are in clinical trials. The goal of DMD NBS is to equitably identify infants with DMD early and initiate treatment earlier, when it is more likely to be effective, to prevent families from experiencing an unnecessary diagnostic odyssey and to ensure that every individual with DMD and their family receives timely, helpful, and accurate resources at the time of diagnosis.

### 1.1. Newborn Screening

In the United States, screening of newborns is under the purview of state public health departments. Each state decides which disorders to screen, and expansion of each state’s panel of screened conditions occurs on a state-by-state basis. The federal government also plays a role in NBS through the Advisory Committee on Heritable Disorders in Newborns and Children (ACHDNC) of the Secretary of Health and Human Services (HHS) [7]. The ACHDNC is charged with evaluating conditions for screening newborns and children for heritable disorders recommending to the Secretary of HHS the conditions for which newborns and children should be screened. If accepted by the Secretary, these conditions become part of the Secretary’s recommended uniform screening panel (RUSP) which is made up of core and secondary conditions. Currently, there are 36 core and 25 secondary conditions on the Secretary’s RUSP. ACHDNC also established a system of nomination and evidence review to evaluate conditions that are candidates for screening [8]. Nomination to the RUSP can be made by an individual or an organization and encompasses a series of reviews to ascertain the nomination’s administrative completeness and scientific and clinical adequacy. If the nomination is accepted and endorsed by the majority of ACHDNC members, a formal evidence review is conducted by a workgroup external to the committee. The external evidence review workgroup reviews the analytical and clinical validity of the proposed screening and diagnostic methods, the predicted health benefits and outcomes, the harms or benefits from screening, and the impact on public health. Key to the nomination of a candidate condition is the complete description of the disease, test, treatment, and completion of a pilot study, with the prospective identification of a newborn subsequently diagnosed with the piloted condition [8].

### 1.2. DMD Pilot

To realize the goal of adding DMD to the RUSP and screening all newborns in the US for DMD, a prospective, population-based pilot that screens both sexes, establishes the analytical and clinical validation of a screening method, and results in at least one DMD diagnosis is needed. Over the past four decades, several pilot studies of NBS for DMD have been completed around the world. These pilot studies tested dried blood spots for creatine kinase (CK) activity, which is elevated in individuals with DMD [9,10]. In addition, during the past 10 years, in the US, there have been several efforts to examine both the evidence to support DMD NBS and the infrastructure needed for screening. In 2012, the National Institutes of Health (NIH) and the Muscular Disease Association (MDA) convened a symposium to examine all available data supporting the implementation of DMD NBS [4]. In 2014, Parent Project Muscular Dystrophy (PPMD) formally initiated a national DMD NBS effort and published a roadmap for DMD NBS [5]. PPMD’s Duchenne NBS efforts have included the expertise and input of experts and leaders within NIH, Health Resources and Services Administration (HRSA), FDA, Centers for Disease Control and Prevention (CDC), American College of Medical Genetics and Genomics (ACMG), the broader NBS community, and the Duchenne community.

In 2018, the Duchenne Newborn Screening Consortium (DMDNBSC, Consortium) was formed and included PPMD, the American College of Genetics and Genomics (ACMG), the New York State Newborn Screening Program (NYS NBS Program), and several commercial companies with an interest in muscular dystrophies and NBS. The consortium created a unique partnership across various stakeholder groups to conduct a two-year, prospective pilot of DMD, while creating a roadmap for future efforts. A key goal of the pilot is to submit a nomination for DMD to be recommended to the RUSP. This paper describes this ground-breaking pilot at the one-year mark, detailing the screening and diagnostic protocols. Clinical follow-up data are not included.

## 2. Materials and Methods

### 2.1. Protocol Development

Successful RUSP addition requires a prospective, population-based pilot that leads to the diagnosis of at least one newborn and the collection of data to support the eventual implementation of screening in both low- and high-birth states [8]. For this NBS pilot, we employed a consortium approach consisting of funding by industry partners, administrative oversight by PPMD, screening by the NYS NBS Program, and data coordination and collection by ACMG’s Newborn Screening Translational Research Network (NBSTRN) [11]. The NBSTRN is a key component of the *Eunice Kennedy Shriver* National Institute of Child Health and Human Development (NICHD) and is operated by ACMG (https://nbstrn.org/, accessed on 27 January 2021). The NBSTRN has developed federally approved secure informatics tools, to support the analytical and clinical validation of new technologies for NBS, and has successfully coordinated the pilots of conditions added to the RUSP. Industry partners included PTC Therapeutics, Sarepta Therapeutics, PerkinElmer, Solid Biosciences, Wave Life Sciences, and Pfizer Inc.

The pilot protocol included: (1) evaluation of a laboratory method to screen for DMD; (2) evaluation of molecular testing to confirm a DMD diagnosis; (3) referral to clinical care; (4) longitudinal collection of clinical data on diagnosed newborns; and (5) developing infrastructure for educating parents and health-care providers about DMD NBS (Box 1). The pilot protocol received Institutional Review Board (IRB) approvals from the New York State Department of Health and the two hospital networks—Northwell Health and New York-Presbyterian (NYP)—included in the pilot (Institutional Review Board Statement). Data-sharing policies and procedures were established and formalized through agreements.

Box 1Goals of the pilot.Validate a high-throughput first-tier immunoassay screen for Duchenne in a high-birth-number state and determine the utility of the Collaborative Laboratory Integrated Reports (CLIR) tool and possible other biochemical markers for interpreting results.Optimize a second-tier molecular testing strategy for confirming diagnosis of DMD and other muscular disorders, after positive CM-MM first-tier screening.Using the first- and second-tier testing algorithm, identify infants who will develop DMD before clinically detectable symptom onset and enable parents the opportunity to see a subspecialist to confirm the diagnosis, identify the DMD genotype, and determine treatment course, including participation in clinical trials.Use the results of the pilot testing to provide evidence required for state and federal assessments of the benefits and risks of NBS for Duchenne and develop the infrastructure to educate parents and health care providers about Duchenne NBS.Nomination of DMD to the RUSP.

### 2.2. Enrollment

Enrollment began in October 2019, using two NYS hospital systems: Northwell Health (Long Island Jewish Medical Center, North Shore University Hospital, Lenox Hill Hospital, and Southside Hospital) and NYP (Weill Cornell Medical Center, Children’s Hospital of New York-Presbyterian, Allen Hospital, Lower Manhattan Hospital, and New York-Presbyterian/Queens). The two hospital systems have the largest number of births in the state (>40,000 in 2020) [12]. Parents or legal guardians of infants born in the hospital systems were invited to participate in the pilot and were provided with a brief verbal explanation of the study, a copy of the study brochure, and a link to a short video explaining the study. At least one parent or legal guardian of the baby provided consent for their newborn to be enrolled in the pilot.

As the pilot progressed, modifications to the study protocol were made, and were submitted, reviewed, and approved by the relevant IRBs. For example, because of the COVID-19 pandemic, from March 2020 to approximately July 2021 (dates varied depending on hospital location), study staff worked remotely, and, therefore, consent to participate in the study was obtained remotely over the phone or by email. During this time, parents or guardians who could not be contacted during their hospital stay were contacted for recruitment usually within four weeks after the infant’s discharge from the hospital, after the routine NBS panel was completed. Recruitment was conducted by phone or phone followed by email, when emails were available. Up to three voicemails and emails were sent by the same pilot personnel in the preferred language identified in the electronic medical record system.

Parents who provided verbal consent by phone were emailed the link to an electronic consent in their preferred language and were given the option to stay on the line with the pilot personnel so they could assist participants whilst completing the electronic consent. Parents who were not able to sign an electronic consent were mailed a paper consent with a stamped addressed return envelope with personnel contact instructions if questions arose. Additionally, protocol changes were made for the confirmatory second-tier genetic testing, to allow for telehealth genetic counseling visits and for sending sample-collection kits to the patient’s residence. Remote sample collection was performed by the parent or guardian in their residence using a buccal swab to collect buccal cells from their infant [13].

### 2.3. Screening

The routinely collected Guthrie cards were used for screening. The NYS NBS Program is the first US state NBS program to use the FDA-authorized PerkinElmer GSP Neonatal Creatine Kinase-MM kit to measure creatine kinase isoenzyme specific for skeletal muscle (CK-MM) for a pilot study [14]. The kit is used in combination with the PerkinElmer GSP analyzer, a high-throughput biochemical analyzer. Forty-one state NBS programs currently use GSP instrumentation within their NBS programs to screen for other conditions [15]. At the beginning of the pilot in October 2019, the GSP Neonatal Creatine Kinase-MM kit was not yet an FDA-authorized kit. Prior to employment of the kit for the pilot, the NYS NBS program performed an extensive validation of the kit for use in the pilot study, as required by New York State. The validation material was submitted to the NYS Clinical Laboratory Evaluation Program (CLEP) for review and approval, as is routine for any non-FDA-authorized test in NYS. Following approval of the validation data by CLEP, the NYS NBS program began testing consented newborns using the CK-MM kit. Each day, a 96-well plate including a calibration curve with six standards was punched in duplicate, and three kit quality control (QC) materials were run in duplicate. The kit received FDA authorization in December 2019.

Plates with patient specimens were run within 24 h of the calibration curve, in accordance with the instructions of the manufacturer. Each patient plate also included three wells with kit QC material. During validation studies for the GSP Neonatal CK-MM kit, it was apparent that CK-MM values were high at birth, especially within the first three days of life, and decreased with age. The NYS NBS program, therefore, selected four different referral cut-offs for four categories of newborns based on the age at specimen collection: (1) 0–47 h, (2) 48–71 h, (3) 72 to 167 h, and (4) ≥168 h. The referral cut-offs decreased with age at collection [16]. No other variables (e.g., sex, gestation age, race, etc.) were used to determine cut-off values and normal ranges.

The screening strategy is illustrated in Figure 1. Any sample with a value below the cut-off was reported as screen negative. Any specimen with a value above the cut-off level was retested in duplicate, and the average of the values was used to determine which follow-up action was required. Any specimen with an average value in the borderline category was reported as “borderline”, and a repeat NBS specimen was requested. A borderline result is considered slightly elevated but below referral level. In these circumstances, the NBS program would request a repeat NBS specimen be submitted for repeat testing by the study sites. In the majority of cases with borderline results, the CK-MM value in the repeat specimen normalized. Any sample with an average above the referral cut-off was considered screen-positive and was referred for molecular testing and to the designated specialist at a Northwell Health or NYP hospital.

CK-MM results were reported along with routine NBS panel results and were made available to the pediatrician, the birth hospital, and the DMD study coordinators. During the pandemic, some families consented to the study while the routine NBS was in progress or had been completed. In these cases, an amended report that included the DMD results was generated and released after the routine NBS report. In the case of a referral, a geneticist or genetic counselor contacted the family and pediatrician and provided information regarding positive DMD screening results and follow-up genetic testing. Parents were offered genetic counseling as part of the study and were asked to provide consent for molecular testing. Box 2 details the molecular testing algorithms when a second specimen (either using whole blood or cells from a buccal swab) was collected for deletion/duplication and sequence variant analysis of the dystrophin encoding *DMD* gene using next-generation sequencing (NGS) analysis. If no pathogenic/likely pathogenic (P/LP) variants or variants of uncertain significance (VUS) were detected, a panel with an additional 45 genes associated with neuromuscular disorders was performed. The order of the del/dup analysis and NGS of up to 47 genes was reversed for females, per the protocol of the molecular genetic laboratory. For a subset of newborns born at Northwell Health Hospitals, if no variants were identified, an additional gene panel with up to 151 genes associated with neuromuscular disorders was performed. Parents were seen by the genetic counselor and/or geneticist for post-test genetic counseling, either in person or via a telehealth visit [17]. Genetic testing results were uploaded into the patient’s electronic medical record and reported to the pediatrician, the NYS NBS program, and NBSTRN.

Box 2Molecular Testing Algorithm.*DMD Sequencing and Microarray-based Comparative Genomic Hybridization (aCGH) Analysis*: In solution hybridization of the 79 coding exons, the muscle promoter as well as the region surrounding several known deep intronic pathogenic variants, within the *DMD* gene. Direct sequencing of the amplified captured regions performed using next generation short base pair read sequencing. A custom aCGH for the *DMD* gene was used to detect deletions and/or duplications.*If needed, perform neuromuscular disorders panel (47 genes):* In solution hybridization of the targeted coding exons within the genes tested.* The genes on this panel were chosen through evidence-based analysis and direct sequencing of the amplified captured regions was performed using next generation short base pair read sequencing.*If needed, perform additional analysis (90 to 104 genes):* These gene panels include sequencing and deletion/duplication testing by NGS of up to 103 additional genes associated with neuromuscular disorders and related neurological disorders.****ACTA1, AMPD1, ANO5, CAPN3, CAV3, COL6A1, COL6A2, COL6A3, CRPPA, DES, DMD, DYSF, EMD, FKRP, FKTN, GAA, GNE, ISPD, ITGA7, LAMA2, LARGE1, LMNA, MYOT, NEB, PLEC, PMM2, POMGNT1, POMT1, POMT2, PYGM, RYR1, RYR2, SELENON, SGCA, SGCB, SGCD, SGCE, SGCG, SIL1, TCAP, TNNI2, TNNT1, TPM2, TPM3, TRIM32, TTN, VCP****ADSSL1, AGRN, ALG14, ALG2, ATP2A1, B3GALNT2, B4GAT1, BAG3, BIN1, CACNA1S, CASQ1, CCDC78, CFL2, CHAT, CHKB, CHRNA1, CHRNB1, CHRND, CHRNE, CLCN1, CNTN1, COL12A1, COL13A1, COLQ, CPT2, CRYAB, DAG1, DNAJB6, DNM2, DOK7, DPAGT1, DPM1, DPM2, DPM3, FHL1, FKBP14, FLNC, GFPT1, GMPPB, GOSR2, GYG1, GYS1, HACD1, HNRNPA2B1, HNRNPDL, ISCU, KBTBD13, KCNJ2, KLHL40, KLHL41, KLHL9, LAMB2, LAMP2, LDB3, LIMS2, LMOD3, LRP4, MAP3JK20, MATR3, MEGF10, MICU1, MTM1, MTMR14, MUSK, MYH2, MYH7, MYL2, MYO18B, MYPN, ORAI1, PNPLA2, POMGNT2, POMK, PREPL, PYROXD1, RAPSN, RXYLT1, SCN4A, SLC18A3, SLC5A7, SMCHD1, SMN1, SMN2, SNAP25, SPEG, SQSTM1, STAC3, STIM1, SUN1, SUN2, SYNE1, SYNE2, SYT1, TAZ, TIA1, TK2, TMEM43, TNNT3, TNPO3, TOR1AIP1, TRAPPC11, TTN, VAMP1, VMA21*

Three surveys were developed to gather information from parents of babies who had a positive CK-MM screen. The surveys were used as a follow-up instrument to address the issues of critical importance to NBS for DMD: evaluation of the informed consent process, factors influencing loss to follow-up, acceptability of the screening to parents, impact on parents receiving a positive CK-MM screen, and parental attitude concerning early diagnosis. The information gathered from parents regarding their experiences will aid in making future improvements in NBS and follow-up for DMD. Survey results will be reported in a subsequent publication.

### 2.4. Diagnosis and Referral to Clinical Care

Results of DNA analysis were disclosed by a geneticist or genetic counselor during a post-testing genetic counseling visit, which was in person, via telehealth, or telephoned to the parents or guardians [17]. In cases with diagnostic molecular testing for DMD or another muscular dystrophy, parents/guardians of the affected infant were referred to a neuromuscular clinic and provided with information on all available treatment options, including the standard of care and any clinical trials for which the baby might be eligible, as defined by approved clinical protocols. Children with unexplained elevated CK were offered follow-up genetic counseling, additional genetic testing, repeat evaluation of the CK, neurological evaluation, and parental testing of any variants of uncertain significance, when appropriate.

To guide the newborn’s health care provider, ACMG developed a clinical-decision support tool called ACTion sheets (ACT Sheets) [18]. ACT Sheets are a web-based resource on the ACMG website designed to guide physicians through preliminary responses to out-of-range NBS results. They contain a summary of differential diagnoses, descriptions of the condition, actions to be taken by the provider, diagnostic evaluation, clinical considerations, reporting requirements, and links to additional resources. ACT Sheets are designed to be supplemented by state-specific information regarding referral resources [18,19,20,21].

Mothers of the infants with a P/LP variant in the *DMD* gene identified through the screening are at risk to be carriers. The exact risk depends upon family history, with approximately 30% of boys with Duchenne having a *de novo* variant. This risk was explained by a geneticist or genetic counselor, and a letter summarizing the information was provided. Carrier testing is readily available. Female carriers are generally asymptomatic during infancy, although some may have musculoskeletal symptoms and/or cardiac manifestations later in life [22]. Symptomatic carriers are more likely to have elevated CK levels [23,24]. Since any female infants identified via NBS have significantly elevated CK-MM, they may be at higher risk to develop symptoms. The American Academy of Pediatrics (AAP) recommends that all carriers should have a cardiac evaluation, including electrocardiogram (EKG) and echocardiogram testing, at least every five years, starting at age 25. Furthermore, detecting a female carrier may assist in detecting DMD in other individuals within the family who have not yet been screened for DMD (e.g., mothers, brothers, cousins, or other females in the family) [25]. Testing of additional family members was performed through usual clinical practice.

### 2.5. Data Collection

The pilot used the NBSTRN’s Longitudinal Pediatric Data Resource (LPDR) to collect, analyze, visualize, and report screening, diagnostic, and health information [26]. The LPDR is a federally approved information technology system that provides a secure environment for genomic and clinical data. We formed a workgroup of experts in Duchenne and other neuromuscular disorders to design a series of questions and answer choices to document the clinical course and assess health outcomes in diagnosed newborns. The expert workgroup drafted question and answer choices as common data elements (CDEs), and NBSTRN created Research Electronic Data Capture (REDCap) case report forms using the CDEs [27,28]. The CDEs were deposited within the NIH Common Data Elements (CDE) Repository to facilitate future efforts and support national data standardization [29].

Data entry into a local REDCap environment occurred at each participating institution, with monthly uploads of de-identified, case-level data into the LPDR. The case-level data on enrolled newborns included race, ethnicity, and type of health insurance, which served as an agreed-upon surrogate measure of socioeconomic status. In addition, demographic and clinical data related to NBS was entered into the NYS NBS Laboratory Information Management System. These data were de-identified and shared with NBSTRN. Obstetric and birth history, including birth anthropometric parameters and gestational age at birth, were also collected. Longitudinal health information was abstracted from the medical records of diagnosed infants and shared with NBSTRN. The pilot also included a parent survey to gather feedback on the pilot process. NBSTRN created and disseminated data dashboards of key parameters in the pilot, including the total number of enrolled subjects per month and the percentage of males versus females (Figure 2).

## 3. Results

### 3.1. Enrolled

In year one of the NYS DMD pilot, a total of 15,793 newborns were enrolled (Northwell Health, *n* = 9231, and NYP, *n* = 6562), while specimens from 15,754 newborns were received and screened at the DOH NBS program for an uptake of 85% (Table 1). The number of enrolled newborns is larger than the number of screened due to the length of time between the process of enrollment and laboratory screening, and the following results are based on the number screened. In total, 51% (8009/15,754) of screened newborns were identified as males and 49% (7689/15,754) were identified as females, and < 1% (56/15,754) were reported as having ambiguous genitalia or the information was unknown. In total, 87% (13,758/15,754) were of average birth weight (2500 to <4000 g) and 87% (13,736/15,754) had a gestational age between 37 0/7 and 40 6/7 weeks. Additionally, 59.9% (9430/15,754) had gestation from 39 0/7 to 40 6/7 weeks [30,31]. Following CK-MM testing, 99% (15,599/15,754) had a CK-MM value within normal limits and were categorized as screen-negative.

### 3.2. Borderline

Less than 1% (140/15,754, 1 in 113) of enrolled babies had a borderline CK-MM value, so a repeat specimen was requested. One baby with a borderline result was subsequently referred based on the elevated CK-MM result on the repeat screen.

### 3.3. Referred

Overall, 16 out of 15,754 newborns (1 in 985) had a CK-MM value above the referral cut-off, and they were referred for molecular testing and clinical follow-up. Forty-four percent (7/16) were male, and fifty-six percent (9/16) were female (Table 2). The average age at the time of collection of the initial sample for referred patients was 28.2 h, with a range of 24 to 58 h. Nineteen percent (3/16) of the infants weighed less than 2500 g and were considered to be low birthweight. Of the total babies screened, 6.5% weighed less than 2500 g. No infants were considered high birthweight (≥4000 g). CK-MM values for referred infants ranged from 4150 to 18,574 ng/mL.

### 3.4. DMD Diagnosed

A newborn was considered diagnosed with dystrophinopathy when they had elevated CK-MM and a P/LP variant in the *DMD* gene diagnostic of DMD or BMD. Three babies met this definition, and all were male (subjects 1, 2, and 3 in Table 2). Two of the three babies had a reported family history of a dystrophinopathy (subjects 2 and 3), and the mother of one of these babies was a known carrier (subject 3). This baby was referred to the MDA clinic based on the CK-MM elevation and the mother’s carrier status, so the baby is not counted in the population prevalence for this first year of our two-year pilot. The second baby’s mother was an obligate carrier, and the third baby’s mother was not known to be a carrier but was identified as a carrier through cascade screening. All three babies are followed in the MDA clinic, and longitudinal health information is being collected for two years following diagnosis. An update on the longitudinal follow-up findings will be included in future reports, after the conclusion of the two-year pilot.

### 3.5. Non-DMD Screen Positive Cases

Ten of the remaining thirteen screen-positive babies without a DMD diagnosis had expanded neuromuscular panel testing performed. Two babies had pathogenic variants identified through this expanded testing but neither were diagnostic. One baby had a pathogenic variant in the *GNE* gene. This gene can be associated with autosomal dominant sialuria disease; however, the variant identified in this child is not typical of sialuria. Upon clinical evaluation, the baby had no evidence of sialuria and was released from follow-up care. Another baby had a pathogenic variant in the *SGCA* gene associated with autosomal recessive limb girdle muscular dystrophy. No second pathogenic variant was identified, and the child was released from clinical follow-up care. All babies who had an expanded neuromuscular panel had one or more VUSs identified. None of the VUSs required follow-up evaluation and were determined unrelated because of inheritance or characteristics of associated diseases.

Two babies were referred for clinical evaluation unrelated to the study. One baby had elevated liver transaminases and congenital heart disease and was diagnosed with Alagille syndrome through clinical molecular testing. Another baby was evaluated by metabolic genetics because of concern for an inborn error of metabolism on the standard NBS, and all follow-up metabolic testing was negative. Exome (trio) with mitochondrial genome analysis was also performed, and it was negative.

CK-MM levels normalized in 6 of the 13 babies (including the child with Alagille syndrome). Eight babies had normal development, as reported by the parent or clinician at last point of follow-up (1 month to 9 months), including four babies with normalized CK. Three screen-positive babies were lost to follow-up, including the two babies who declined molecular testing and the one baby who moved out of state shortly after birth (Table 2). Overall screening results are presented in Figure 3.

## 4. Discussion

At the one-year mark of this two-year prospective pilot executed across a diverse population, findings were used to support the implementation of NBS for DMD. In the first year, we successfully validated a screening method, optimized a second-tier NGS method for confirming DMD and other muscular dystrophies, diagnosed three male infants with DMD/BMD, and collected evidence to support a RUSP nomination. The first year was conducted in the midst of a worldwide pandemic, which impacted all aspects of the pilot and informed changes to the protocol that were implemented in the second year [32]. While none of the identified DMD variants in the three diagnosed babies were amenable to current FDA-authorized treatments, the identification of a *DMD* pathogenic variant in infancy allows for genetic testing for at-risk mothers, siblings, and extended relatives and enables family-planning options such as prenatal testing or preimplantation genetic testing for future pregnancies. Early diagnosis and knowing the variant may enable treatment with mutation-specific therapy in the future.

The mean age at first signs and symptoms of DMD as reported by the caregiver are 2.7 (Standard Deviation (SD) = 1.8, Median (Md) = 2.0) years for males, with the average time from first symptoms to diagnostic confirmation being 2.2 (SD = 2.5, Md = 1.4) years [33]. DMD has traditionally been treated with corticosteroids that demonstrate anti-inflammatory and immunosuppressant properties. Corticosteroids are not typically administered until four years of age, when irreversible damage may have already occurred. Currently, FDA-authorized treatments include exon-skipping therapies for exons 45, 51, and 53, which are likely to be most effective when initiated early before irreversible muscle injury. Such therapies aim to shift out-of-frame deletions to in-frame deletions, hopefully resulting in a smaller-yet-functional protein manifesting as a milder clinical presentation. Along with the available FDA-authorized treatments, the DMD drug-development pipeline features additional exon-skipping therapies, nonsense mutation readthroughs, as well as gene therapies, which introduce a miniature version of the dystrophin that can rescue muscular function likened to Becker muscular dystrophy, progressing through phase trials for FDA authorization [6,34,35,36,37,38,39].

The psychological impact on parents on the time to diagnosis has been a focus on the efficacy on NBS for DMD. The results from the first year of this pilot show the potential for NBS programs to diagnose and, thus, provide early treatment or intervention that will slow disease progression and, even, lead to improved longitudinal health outcomes. While the CK-MM kit proved to be an effective screen for diagnosing DMD, the pilot demonstrated that the potential for identifying false-positives through the screening process can cause parental stress. Increasing educational resources focused on data utilization to explain transient biomarker elevations has the potential to increase medical literacy and enhance parental trust within the medical system. Additionally, timely diagnosis and intervention will aid parents in their navigation of the previously challenging diagnostic odyssey experienced by families associated with obtaining a DMD diagnosis [40,41].

## 5. Conclusions

The first year of the pilot was able to demonstrate the feasibility of screening for DMD through identification of three affected males. In addition to diagnosing DMD, outcomes provide opportunity to improve upon the pilot protocol and inform standard of care to ensure optimal follow-up care for babies and support for their families. 

During the pilot’s second year, we will work to streamline the second-tier molecular testing for referred infants, including strategies for managing VUS and exploring the feasibility of conducting NGS using dried blood spots. We will also continue to collect longitudinal health information to inform the medical management of newborns who do not have a clear diagnosis and analyze the full dataset to define norms of CK-MM values by gestational age, informing the interpretation of screening results for premature infants. The data capture and analysis of the parent surveys that were created during year one help us to understand the newborns who were lost to follow-up (LTFU), the decision-making of parents who refused additional screening and/or testing, and attitudes about the acceptability of screening and the benefit of early diagnosis. These analyses will be completed at the end of the pilot. Collectively, these data will provide a useful roadmap for DMD priorities, including the RUSP nomination and the eventual nationwide implementation of newborn screening for DMD, and will also guide the design of pilots for other genetic conditions that are candidates for newborn screening.

## Figures and Tables

**Figure 1 IJNS-08-00050-f001:**
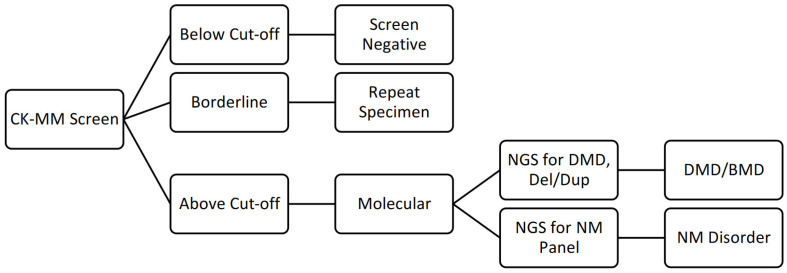
Screening algorithm used in the DMD pilot study. NGS: next-generation sequencing; Del/Dup: deletion/duplication; NM: neuromuscular.

**Figure 2 IJNS-08-00050-f002:**
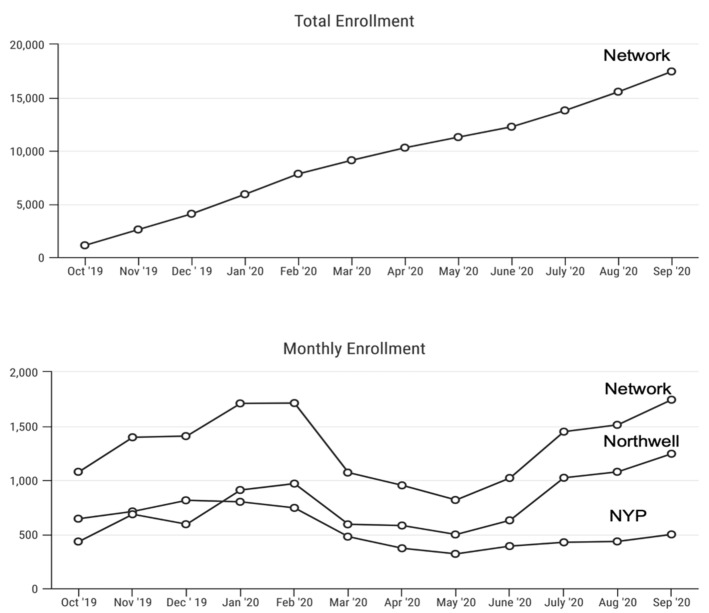
Data dashboard: NBSTRN created a monthly dashboard that included monthly and total enrollment numbers for Northwell, NYP, and overall network.

**Figure 3 IJNS-08-00050-f003:**
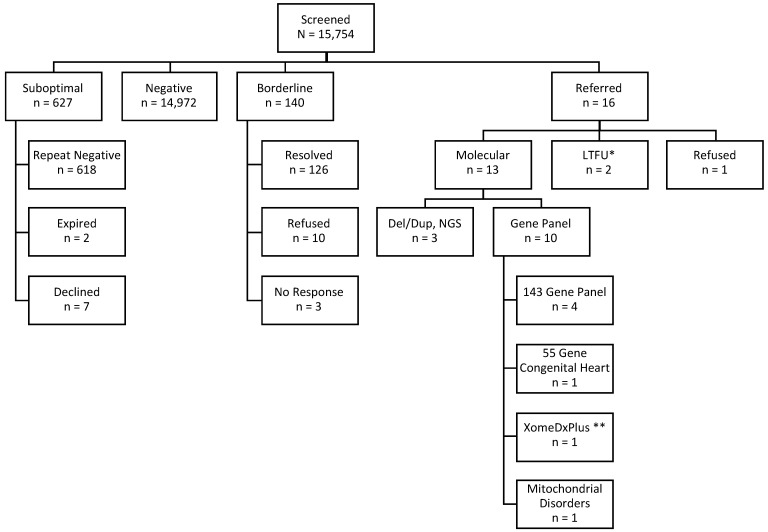
Overall screening results—* LTFU = lost to follow-up. ** XomeDxPlus is clinical exome sequencing with mitochondrial genome sequencing/deletion testing.

**Table 1 IJNS-08-00050-t001:** Year one enrollment summary (October 2019 to September 2020).

Total Hospital Births	35,570		
Missed	16,980 (48%)		
Approached	18,590 (52%)	Declined to Participate	2797 (15%)
		Enrolled	15,793 (85%)

**Table 2 IJNS-08-00050-t002:** Characteristics and results for newborns referred to molecular testing organized by resolution.

Case *	Diagnosis	Sex	Screen Result CK-MM (ng/mL)	Molecular Result	Case Summary
R-5	DMD	M	7809	*DMD* dup ex18 (LP)	Molecular diagnosis of Duchenne/Becker muscular dystrophy. Followed in MDA clinic. Last visit at 17 months: out-toes but no toe walking. With support can raise self-off floor. Cannot assess if Gower sign is present. Appears to have trouble raising body with use of one leg. Physical therapy 2x per week and development therapy 1x per week. No medications.
R-6	BMD	M	6384	*DMD* del ex48-49 (LP)	Followed in MDA clinic. No cognitive or motor delays identified at 7 months. Sequencing result is consistent with a deletion of this region of the gene and predicted to result in in-frame deletion in the DMD mRNA. This is consistent with a diagnosis of DMD/BMD. The subject’s maternal grandfather was diagnosed with molecularly confirmed BMD. His symptoms started in mid-life.
R-15	DMD	M	18,574	*DMD* del ex3-43 (LP)	Not screened as a newborn. Referred to genetics and enrolled in study because of family history of DMD, markedly elevated CK, and troponin T suggestive of congenital myopathy and hydronephrosis. DMD and gene panel ordered concurrently.
R-8	Alagille Syndrome	F	4370	*JAG1* (P)	CK normalized at 4 months. Baby had a congenital heart defect and very high liver enzyme. Clinical genetics evaluation revealed *JAG1* pathogenic variant consistent with Alagille syndrome.
R-12	None	F	4150	*SGCA*, (P), het; *TTN* (VUS), het	Normalized CK and no evidence of muscle weakness at 1 month. Family received genetic counseling regarding AR inheritance of limb girdle muscular dystrophy and was offered parental testing.
R-9	None	F	6007	*GNE* c.218G>A, (P), het; *POMT1* (VUS), het; *TTN* (VUS) x 3, het	CK not repeated. Clinical evaluation showed no evidence of sialuria.
R-3	None	F	4895	*DMD* 17 kb del intron 55 (VUS), het; *RYR1* (VUS), het	Normalized CK and no evidence of weakness at 9 months.
R-10	None	F	5365	*LAMA2*, (VUS), het	CK not repeated. Declined further follow-up care.
R-11	None	M	4154	*TTN*, (VUS), het	Normalized CK and no evidence of weakness at 1 month.
R-13	None	M	5128	*SIL1*, (VUS), het	CK not repeated. Declined further follow-up care.
R-14	None	F	12,002	*AMPD1*, (VUS), het	CK was not repeated. Family moved out of state.
R-16	None	F	4507	*RYR2*, (VUS), het; *TTN*, (VUS), het	CK normalized at 9 days. Declined further follow-up care.
R-4	None	M	4850	*DYSP*, (VUS), het *PLEC*, (VUS), het *RYR2*, (VUS), het	Normalized CK at 10 days. At birth there was a concern for inborn error of metabolism because of the standard newborn screening panel. Complete metabolic workup and exome sequencing with mitochondrial genome seq/del were negative.
R-7	None	F	5054	Declined testing	Parents report normal development at 7 months. Declined molecular testing.
R-1	LTFU	M	4593	NA	Unknown.
R-2	LTFU	F	8399	NA	Unknown.

* Case numbers derived from the referral to molecular testing order e.g., the first case referred for molecular testing is R-1. F = female; M = M=male; LP = likely pathogenic; VUS = variant of unknown significance; P = pathogenic; NA = not applicable; CK = creatine kinase; *DMD* = dystrophin; *JAG1* = jagged canonical notch ligand 1; *SGCA* = sarcoglycan alpha; *TTN* = titin; *GNE* = glucosamine (UDP-N-acetyl)-1-epimerase/N-acetylmannosamine kinase; *RYR1* = ryanodine receptor 1; *LAMA2* = laminin subunit alpha 2; *SIL1* = SIL1 nucleotide exchange factor; *AMPD1* = adenosine monophosphate deaminase 1; *RTR2* = putative protein serine/threonine phosphatase; LTFU = lost to follow-up.

## Data Availability

The data presented in this study are available upon request from the corresponding author.

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
