# Peer review of "Newborn Screening for Duchenne Muscular Dystrophy: First Year Results of a Population-Based Pilot"

_2409-515X, 2022, doi:10.3390/ijns8040050_

Round 1
Reviewer 1 Report
The authors present the result of an interesting pilot of NBS for DMD conducted in the States of New-York, with the aim of including DMD in the RUSP. The paper is easy to read and understand.
Minor points of form:
- NBSTRN : do not use abbreviations in the abstract
- NYS DOH : abbreviation not explain
- 235-236 : double space
I would have liked more information on the methodology followed and on different points:
- Did the parents receive information about the pilot project during the pregnancy?
- On which dried blood card was the blood collected? The same as for the standard Guthrie, a specific one? How many drops of blood were needed?
- The collection of consent was done during the Covid period by telephone: was it not possible to have the consent signed online by default? or to ask for a signature on a document sent by e-mail after information by telephone?
- Lines 101-113: the multitude of abbreviations makes this paragraph difficult to understand and ultimately of little interest to the project as a whole.
- Need for more details on the kit used: line 146: state that they are using a validated kit, and a few lines later (153), this kit was not yet validated. Wording to be changed.
- Line 195: what other neuromuscular diseases were screened? At the same time as for DMD? Why is this not presented? It can be understood by all (including parents) that it is only about screening for DMD. One learns almost "incidentally" those other diseases are screened. And without knowing which ones. How can an ethical committee allow such vagueness to pass?
- Line 217: A genetic result by telephone seems dangerous: already the diagnosis of an untreatable disease is more than ethically questionable. I understand that in Covid times it was difficult to consult in person, but it should be made clear how these consultations were organised.
A central question remains for me, more on the overall aspect of the project.
It seems essential to me to develop newborn screening for a range of treatable diseases. Developing the methodology for screening for DMD seems positive. However, it seems wrong to present screening for DMD as necessary at present. No treatment is currently approved, let alone at a presymptomatic stage. If this avoids a long diagnostic process and possibly the birth of a second child, it does not fit into the criteria for newborn screening. Normally, the criteria for newborn screening are very strict, based on the Wilson and Jungner concepts, requiring approved treatment for patients with the identified disease. And should only be of direct benefit to the child concerned (not the parents / not the siblings / not the rest of the extended family). It is often argued that early diagnosis without treatment leads to the child being regarded differently and with significant psychological sequelae. The screening of girls also raises questions for the same reasons.
The very purpose of the programme does not seem absolutely necessary, as noted in line 76-77: "Over the past four decades, several pilot studies of NBS for DMD have been completed around the world. "But these pilot projects have all been discontinued because they did not meet the needs. What is the need for a pilot project if it does not have the ultimate goal of being transformed into an official programme that can be used by all? As the methodology is already validated, the implementation of this screening can be easily done when the treatments arrive. We all hope this will be soon, but probably not for a few years. So what is the benefit for the children screened so far?
More questioning and arguing about the purpose of the programme itself would greatly enrich the discussion.
Author Response
Thank you very much for your thoughtful and helpful review of this manuscript. We made the updates as suggested. Please see below for an item-by-item response.
I would have liked more information on the methodology followed and on different points:
- Did the parents receive information about the pilot project during the pregnancy?
The parents did not receive information about the pilot project during the pregnancy. Families were only recruited after birth.
- On which dried blood card was the blood collected? The same as for the standard Guthrie, a specific one? How many drops of blood were needed?
The dried bloodspot was the same standard Guthrie card collected for newborn screening. As per the kit’s instructions, a 3 mm DBS was used for testing.
- The collection of consent was done during the Covid period by telephone: was it not possible to have the consent signed online by default? or to ask for a signature on a document sent by e-mail after information by telephone?
All consents were obtained online. It was important to the principal investigators to call participants to explain the study Consents are complicated and for the community, that includes low literacy/low medical literacy, and it was determined that a telephone call explaining the study increased meaningful consent.
- Lines 101-113: the multitude of abbreviations makes this paragraph difficult to understand and ultimately of little interest to the project as a whole.
The comment is greatly appreciated for clarity for our audience. It has been fixed for more clarification.
- Need for more details on the kit used: line 146: state that they are using a validated kit, and a few lines later (153), this kit was not yet validated. Wording to be changed.
Thank you for this comment as the previous iteration was admittedly difficult to understand. The wording has been changed for clarification beginning on line 164. The pilot study started in October 2019 before the kit was FDA-authorized in December 2019. Therefore, an extensive validation was done to enable use of the kit prior to FDA-authorization.
- Line 195: what other neuromuscular diseases were screened? At the same time as for DMD? Why is this not presented? It can be understood by all (including parents) that it is only about screening for DMD. One learns almost "incidentally" those other diseases are screened. And without knowing which ones. How can an ethical committee allow such vagueness to pass?
The goal of the pilot was to screen for DMD, which is the most common cause of persistently elevated CK-MM in a newborn. However, CK-MM is non-specific, and elevations can be seen in other neuromuscular disorders. As stated, in cases of elevated CK-MM, if DMD genetic testing was negative, additional neuromuscular genes were screened as standard of care. Parents were consented at the time of enrollment about the potential diagnosis of other neuromuscular disorders, and, again, prior to genetic testing for a panel of neuromuscular genes. Because these were all diseases with similar health implications as DMD, parents thought about these conditions similarly.
- - Line 217: A genetic result by telephone seems dangerous: already the diagnosis of an untreatable disease is more than ethically questionable. I understand that in Covid times it was difficult to consult in person, but it should be made clear how these consultations were organised.
With COVID-19, parents were unable/unwilling to come for in person visit so we had to adjust to include telephone or video visits. With video visits personnel was able to see facial expressions and body language, much as in person. There was an experienced team of medical geneticists and genetic counselors returning results who have extensive experience with returning results clinically and as part of research studies. There were multiple interactions with parents for whom positive results were returned to refer babies to neuromuscular specialists, assistant with parental and extended family member genetic testing, and follow up for outcomes. With each touchpoint the pilot personnel were able to determine how parents were coping with the information and support them through their journey. Many of these study patients received greater support over a longer period than clinical patients who receive results in person. While it may seem dangerous and of valid ethical concern, we have included a recent publication describing the current practice of telehealth and e-consultations.
Vora, N. L., Hardisty, E., Coviello, E., & Stuebe, A. (2020). Telehealth to provide prenatal genetics services: Feasibility and importance revealed during global pandemic. Prenatal Diagnosis, 40(8), 1040–1041. https://doi.org/10.1002/pd.5716
A central question remains for me, more on the overall aspect of the project.
It seems essential to me to develop newborn screening for a range of treatable diseases. Developing the methodology for screening for DMD seems positive. However, it seems wrong to present screening for DMD as necessary at present. No treatment is currently approved, let alone at a presymptomatic stage. If this avoids a long diagnostic process and possibly the birth of a second child, it does not fit into the criteria for newborn screening. Normally, the criteria for newborn screening are very strict, based on the Wilson and Jungner concepts, requiring approved treatment for patients with the identified disease. And should only be of direct benefit to the child concerned (not the parents / not the siblings / not the rest of the extended family). It is often argued that early diagnosis without treatment leads to the child being regarded differently and with significant psychological sequelae. The screening of girls also raises questions for the same reasons.
The reason to pilot newborn screening for DMD is that there are FDA approved treatments for some mutations. The most effective first stage of screening is CPK which cannot be limited to certain mutations, and the study was designed to include any DMD mutation identified by molecular testing. While we did not identify infants with these mutations, that was one of the goals. This was the first pilot in the United States to use CK-MM as a biomarker and was successful in identifying at least one infant with DMD as required per Recommended Uniform Screening Panel nomination. Currently approved therapies include exon skipping therapies with many other potential therapies in a clinical trial or will have clinical trials initiated next year.
The very purpose of the programme does not seem absolutely necessary, as noted in line 76-77: "Over the past four decades, several pilot studies of NBS for DMD have been completed around the world. "But these pilot projects have all been discontinued because they did not meet the needs. What is the need for a pilot project if it does not have the ultimate goal of being transformed into an official programme that can be used by all? As the methodology is already validated, the implementation of this screening can be easily done when the treatments arrive. We all hope this will be soon, but probably not for a few years. So what is the benefit for the children screened so far?
This study was necessary to assess the GSP Neonatal CK-MM kit, determine distribution of CK values in a diverse population of newborns, relationship to other clinical parameters, and determine cutoffs for positives. In addition, the pilot study was necessary to develop and refine the second stage of molecular confirmation. Additionally, this pilot study was conducted to explore and potentially identify what else that could be found in screening for DMD. This was done to identify and inform best clinical practice as NBS is done as standard of care as is the expanded gene panel. NBS allows for early intervention which can slow disease onset and progression.
More questioning and arguing about the purpose of the programme itself would greatly enrich the discussion.
We appreciate this concern and have expanded the introduction and discussion. We hope the justification for this prospective pilot is now clearer.
Reviewer 2 Report
I have only minor criticisms.
Introduction: Brief summary of the drugs now available for DMD patients is appropriate.
Materials & Methods/Results: I would appreciate a more detailed report about high CK values at different days of life. Moreover, specify the different cut-off (below, borderline, above).
Table 1, 2nd sentence: ... DMD and other muscular disorders .... instead of ... DMD and other muscular dystrophies ...
Lines 271-272: ... including the total number of enrolled subjects per month and the percentage of males versus females (Figure 2).
However Fig. 2 does not show males vs females enrolled.
Results. 3.2: when a repeat specimen was performed in babies who had a borderline CK value ?
Babies who had normal CK value after a high or very high value at first examination open the problem of false-positives having a possible severe psychological impact on the families. This point has to be cited and discussed (Vita & Vita, Is it the right time for an infant screening for Duchenne muscular dystrophy? 2020).
Author Response
Thank you very much for the thoughtful and helpful review of our manuscript. Please see our response to your feedback below.
- Introduction: Brief summary of the drugs now available for DMD patients is appropriate.
Thank you for this comment, we have added more on current treatment availability – please see paragraph line 47.
- Materials & Methods/Results: I would appreciate a more detailed report about high CK values at different days of life. Moreover, specify the different cut-off (below, borderline, above).
CK-MM is elevated within the first hour of birth and increases substantially (over 4-fold) to reach a maximum at 25 hours of age. It subsequently decreases during the second day of life and then more gradually during one week of life and stabilizes.
Park, S., Maloney, B., Caggana, M., & Tavakoli, N. P. (2022). Creatine kinase‐MM concentration in dried blood spots from newborns and implications for newborn screening for Duchenne muscular dystrophy. Muscle & Nerve, 65(6), 652–658. https://doi.org/10.1002/mus.27533
- Table 1, 2nd sentence: ... DMD and other muscular disorders .... instead of ... DMD and other muscular dystrophies ...
Thank you for providing this correction. It has been implemented within the manuscript for consistency.
- Lines 271-272: ... including the total number of enrolled subjects per month and the percentage of males versus females (Figure 2). However Fig. 2 does not show males vs females enrolled.
This figure provides a snapshot of the developed data dashboards. Further access may be granted upon stakeholder request.
- 3.2: when a repeat specimen was performed in babies who had a borderline CK value?
The NBS Program requested a repeat specimen for any baby with a borderline CK-MM result. Repeat DBS specimens were collected between 20 hours and 139 days after the specimen that had a borderline CK-MM result and submitted to the Program.
- Babies who had normal CK value after a high or very high value at first examination open the problem of false positives having a possible severe psychological impact on the families. This point has to be cited and discussed (Vita & Vita, Is it the right time for an infant screening for Duchenne muscular dystrophy? 2020).
Thank you for raising this concern as it enriches our discussion for the DMD pilot. We have added a paragraph in the Discussion section beginning with line 426.